# Molecular Markers for Detecting Inflorescence Size of *Brassica oleracea* L. Crops and *B. oleracea* Complex Species (n = 9) Useful for Breeding of Broccoli (*B. oleracea* var. *italica*) and Cauliflower (*B. oleracea* var. *botrytis*)

**DOI:** 10.3390/plants12020407

**Published:** 2023-01-15

**Authors:** Simone Treccarichi, Hajer Ben Ammar, Marwen Amari, Riccardo Cali, Alessandro Tribulato, Ferdinando Branca

**Affiliations:** 1Department of Agriculture, Food and Environment (Di3A), University of Catania, 95131 Catania, Italy; 2Laboratory of Molecular Genetics Immunology and Biotechnology (LR99ES12), Faculty of Sciences of Tunis, University of Tunis El Manar, Campus Universitaire, Tunis 2092, Tunisia

**Keywords:** *Brassica* complex species (n = 9), MADH–box genes, SSRs assay, heterozygosity index, allelic variance, inflorescence morphometric traits

## Abstract

The gene flow from *Brassica oleracea* L. wild relatives to *B. oleracea* vegetable crops have occurred and continue to occur ordinarily in several Mediterranean countries, such as Sicily, representing an important hot spot of diversity for some of them, such as broccoli, cauliflower and kale. For detecting and for exploiting the forgotten alleles lost during the domestication processes of the *B. oleracea* crops, attention has been pointed to the individuation of specific markers for individuating genotypes characterized by hypertrophic inflorescence traits by the marker assisted selection (MAS) during the first plant growing phases after the crosses between broccoli (*B. oleracea* var. *italica*)/cauliflower (*B. oleracea* var. *botrytis*) with *B. oleracea* wild relatives (n = 9), reducing the cultivation and evaluation costs. The desired traits often found in several *B. oleracea* wild relatives are mainly addressed to improve the plant resistance to biotic and abiotic stresses and to increase the organoleptic, nutritive and nutraceutical traits of the products. One of the targeted traits for broccoli and cauliflower breeding is represented by the inflorescences size as is documented by the domestication processes of these two crops. Based on the previous results achieved, the numerical matrix, obtained utilizing five simple sequence repeats (SSRs), was analyzed to assess the relationship among the main inflorescence characteristics and the allelic variation of the SSRs loci analyzed (BoABI1, BoAP1, BoPLD1, BoTHL1 and PBCGSSRBo39), both for the *Brassica oleracea* and *B*. *oleracea* wild relatives (n = 9) accessions set. The main inflorescence morphometric characteristics, such as weight, height, diameter, shape, inflorescence curvature angle and its stem diameter, were registered before the flower anthesis. We analyzed the correlations among the allelic variation of the SSRs primers utilized and the inflorescence morphometric characteristics to individuate genomic regions stimulating the hypertrophy of the reproductive organ. The relationships found explain the diversity among *B. oleracea* crops and the *B. oleracea* complex species (n = 9) for the inflorescence size and structure. The individuated markers allow important time reduction during the breeding programs after crossing wild species for transferring useful biotic and abiotic resistances and organoleptic and nutraceutical traits to the *B. oleracea* crops by MAS.

## 1. Introduction

Molecular markers provide a simple, rapid and non-destructive method for selection by genotyping, which can be utilized at any plant stages for significantly reducing the time, cost and other resources required for breeding programs to develop varieties [1]. Marker-Assisted Selection (MAS) is a technique for identifying and localizing genes associated with the key plant traits in its genome. The goal of plant breeding is to create new varieties that combine several traits defined by the breeder to meet the needs of farmers and consumers. It is also of great interest in programs for the introgression of a gene of interest into an elite variety by the backcross plan. MAS is an effective tool in plant breeding, especially when the target phenotypic traits are laborious or expensive to measure and which can be supported by the molecular markers [2].

*Brassica* is a genus of the dicotyledonous plants belonging to the *Brassicaceae* family, native to Eurasia and the Mediterranean basin, and it includes about forty species [3]. They are generally annual or biennial herbaceous plants, in some case perennial, with cross-shaped flowers characteristic of *Cruciferae* (now *Brassicaceae*). Cultivated species have a very diversified plant morphology, depending on whether they have been domesticated for their leaves, petioles, buds, flowers, roots or seeds. They are grown as vegetables, condiments, oilseeds or medicinal plants. Four species are mainly cultivated with an important role in the human diet: *Brassica oleracea* (various coles), *Brassica nigra* (black mustard), *Brassica napus* (rapeseed, rutabaga) and *Brassica rapa* (turnip, rape, Chinese cabbage) [4]. Among the *Brassica oleracea* crops, cauliflower (*B*. *oleracea* L. var. *botrytis* L.) and broccoli (*B*. *oleracea* L. var. *italica*) are the only two crops grown for their hypertrophic reproductive organs, whereas all the others have constantly modified vegetative organs that represent the products [5]. The domestication process of broccoli and cauliflower probably began since Roman times, however, the similar morphological structure of the edible organs of both crops has repeatedly caused confusion in finding unique descriptions for each of them. The apparent similarity of broccoli and cauliflower inflorescence and the similar hypertrophy of the inflorescence may have influenced the scientific and common names, which are interchangeable in some cases [5,6,7]. *B. oleracea* vegetable products provide a high level of bioactive compounds conferring high antioxidant activity related to the richness of glucosinolates, isothiocyanates and polyphenols contained in the edible portion of the plant [8,9].

Recent DNA analysis using molecular techniques supports a high degree of similarity between Sicilian wild *Brassica* species (n = 9) and *B*. *oleracea* crops [10,11,12,13]. More recently, additional similarities have been observed among *B. oleracea* crops and the *Brassica oleracea* complex species (n = 9) widespread in the Mediterranean basin than have been observed for the Atlantic wild *Brassica* ones [14,15,16,17,18,19].

Molecular markers have proven to be important tools for assessing genetic variation and relationships in plant species above all for organic breeding and for establishing the organic heterogenic materials as described by the EU Directive 848/2018 [20]. Some molecular markers are represented by simple sequence repeats (SSRs), alternately known as microsatellite markers, which have been successfully used for evaluating the genetic variability and for distinguishing among them the nearly related *Brassica* genotypes [21,22,23], because of their codominance and ability to reveal a high number of alleles for locus, resulting in a high degree of reproducibility and variability [24].

The MADS-box genes are involved in plants for the control of all major aspects of their development, such as the differentiation between male and female gametophytes, the development of embryos and seeds, roots, flowers and fruits, and the determination of flowering time [25]. Bowman et al. [26] studied flower development genes, and they found several genes involved, like *apetala 1* and *cauliflower*, also observed in *Arabidopsis thaliana*. These genes are closely related to members of the MADS-box gene family, and a mutant copy of these is present in the *B. oleracea* genome C. Irish and Sussex [27] characterized by a lot of floral morphotypes produced by the homeotic recessive *apetala 1* (*ap1*) mutation in *Arabidopsis* and the homozygote for this mutation which demonstrated low inflorescence affecting the formation of floral buds.

Exercising the simple sequence repeat (SSR) marker BoAP1 advanced a number of alleles which were found in the wild *B. oleracea* complex species (n = 9) rather than in cabbage and cauliflower. *BoAp1-a* locus located in a single genomic region on the chromosome 6 of *B. oleracea* with the additional ones (BoCAL, BoLFY, BoAP1-c, BoREM1) of the MADS-box genes family [28]. Smith and King [29] proposed a genetic model grounded on the segregation of the recessive alleles for BoCAL and BoAP1 candidate genes that showed differences during the plant stage of flower development arrest between broccoli and cauliflower. According to Smith and King’s allelic distribution genetic model, the domestication strategy reduced the allelic diversity by promoting *loci* affecting the arrest of floral development that determined the first inflorescence hypertrophy of broccoli and then, by further selection, the cauliflower’s curd inflorescence morphotype; the Sicilian Purple cauliflower was indicated as an important intermediate of this domestication pathway. The four primers BoAP1, BoABI1, BoPLD1 and BoTHL1 were designed by Tonguç and Griffiths [30] to investigate the genomic DNA for evaluating the genetic similarity among several *Brassica oleracea* cultivars belonging to three varietal groups (broccoli, cauliflower and cabbage). One additional primer PBCGSSRBo39 was designed by Burgess et al. [31] to demonstrate a useful molecular marker for crop improvement that was derived from shotgun sequencing methods.

These five SSR primers (BoAP1, BoABI1, BoPLD1, BoTHL1 and PBCGSSRBo39) were chosen by Branca et al. [32] by opting for them from among others primers, for phylogenetic analysis and for evaluating the genetic similarity among different *B. oleracea* accessions (cauliflower and broccoli) and *B. oleracea* complex species (n = 9), as well as to estimate genetic divergence using the FST statistical parameter, where broccoli cultivars grouped with cauliflower cultivars as expected and wild species showed major genetic differences. Sheng et al. [33] characterized and mapped 91 MADS-box transcription factors able to discern from type I (Mα, Mβ, Mγ) and type II (MIKCC, MIKC*) genetic groups as a consequence of phylogenetic and gene structure analysis: 59 genes were randomly distributed on 9 chromosomes, and 23 were located in 19 scaffolds, while 9 of them were not located due to the lack of information on the NCBI database (Sheng et al., 2019). Treccarichi et al. [20] used the set of markers used by Branca et al. [34] to calculate the genetic diversity among nine accessions of *B. oleracea* crops and *B*. *oleracea* complex species (n = 9) and to evaluate the hypertrophic induction of the curd. The SSRs assay can also be exploited in population genetics to discover allelic variants related to interesting traits, and it could also be a topic for the breeders that can apply it to inherit them in the F2 population [35].

In the present work, the above cited five SSR primers based on the sequences of several MADS-box genes were used to analyze the allelic variation of different Sicilian landraces and hybrids F1 of cauliflower and broccoli, and of some *B. oleracea complex* species (n = 9), for associating them with the inflorescence morphometric traits that have been measured for each accession. The following manuscript aims to identify the most interesting allelic variants to use as organic breeding tool for broccoli MAS.

## 2. Results

### 2.1. Bio-Morphometric Analysis

Based on the bio-morphometric characteristics of the inflorescence analyzed were inflorescence weight (IW), height (IH), diameter (ID1), stem thickness (ID2), shape (IS) and angle of curvature (IA). With regard to IW, it varied among genotype from 1095.8 to 16.65 g, for CVF1.1 and BV, respectively (Table 1). IW showed the highest values in cauliflower F1 hybrids and landraces, followed by broccoli heirlooms and landraces. CWRs showed the lowest value of IW varying from 33.3 to 16.7 g, for BU1 and BV, respectively. With regard to IH, the CWRs group, represented by the accessions BM, BU1, BU2, BU3, BU4, BV, BY1, BY2 showed the highest IH values followed by the lowest IW, ID2, ID1 and IA due to the characteristics of their inflorescence architecture, which is large and thin, with large flower buds, low inflorescence density and low bolting resistance. IH varied for the CWRs group from 14.8 to 27.6 cm, while in cauliflower and broccoli groups, it varied from 7.5 to 22.2 cm for CVF1.4 and BR2, respectively. Concerning ID2, we observed the highest values for cauliflower morphotype varying from 28.6 to 39.8 mm for CV3 and CV1, respectively, while broccoli genotypes exhibited an average value of 3.1 cm varying from 2.6 to 3.8 mm for BR8 and BR1, respectively. The crop wild relatives group showed an average ID2 value of 17.9 mm varying from 9.6 to 22.5 mm for BM and BY1, respectively (Table 1). Concerning ID1 it showed the highest value for the cauliflower group varying from 13.6 to 21.1 cm for CV3 and CV4, respectively, and it varied for the broccoli group from 4.7 to 12.3 cm for BR9 and BRF1.1, respectively (Table 1). IS showed the highest values for broccoli accessions varying from 1.2 to 3.6 for BRF1.1 and BR9, respectively (Table 1).

Broccoli hybrids F1 resemble cauliflower inflorescence and, for this reason, showed a lower CS ratio than for broccoli landraces which are characterized by higher values of IH (Table 1). The cauliflower genotype showed intermediate IS values varying from 0.5 to 1.17 for CVF1.2 and CVF1.7, respectively. The CWRs group showed the lowest IS value, which varied from 0.1 for BU3, BU4, and BV to 0.3 for BM (Table 1). Concerning IA character, it exhibited the highest values for the cauliflower genotypes showing the average value of 102.9°, varying from 85° to 117° for CV3 and CV9, respectively (Table 1). Broccoli genotypes are characterized by a reduced amplitude compared to the cauliflower group, and it varied from 279° to 76° for BR9 and BRF1.1, respectively (Table 1). The CWRs group showed the lowest IA values due to their different simple inflorescence architecture that is slenderer than cauliflower and broccoli genotypes, and it varied from 9.5° to 15° for BU3 and BU2, respectively (Table 1).

### 2.2. Identification of the Best Molecular Marker by the Association between Their Allelic Variants and the Bio-Morphometric Traits

Pearson’s correlation showed a significant correlation among IW and the descriptors ID1, ID2 and IA. On the other hand, the IS descriptor is derived from the ratio between ID1 and IH and is negatively related to ID2 (Table 2). Concerning IA, it was positively correlated to ID1, IW and ID2, respectively (Table 2).

By analysing the correlation among the matrix data of the considered alleles detected and the inflorescence morpho-biometric traits, we individuated the ones that correlated the highest.

The correlation among the molecular markers and the inflorescence descriptors showed a high significant correlation with the allelic variation 155 bp of AP1 (P1), which was correlated negatively with IH and positively with IW, ID1, ID2 and IA (Table 3). With regards to the P2 (BoTHL1), the allelic variant of 157 bp molecular weight was positively correlated with IH and negatively with IW, IA, ID1 and IS, respectively, in decrescent order (Table 3). The allele of 165 bp found for P2 was positively correlated with ID1, ID2 and IA, while it was negatively correlated with IW. The allelic variation of 184 bp detected for the marker BoAB1 (P3) was significantly negatively correlated with IS, ID1, WI and IA, and positively with IH (Table 3). The allelic variant of 288 bp of the BoPLD1 marker (P4) was positively correlated with IA, ID1 and IW, and negatively with IH, whereas on the other hand, the allelic variant of 291 bp was positively correlated to IA and IH, and negatively correlated to ID1, IW and ID2, respectively, in decrescent order (Table 3). The PBCGSSRBo39 (P5) allele of 294 bp was positively correlated with IH and IA, and negatively correlated with IW, ID1 and IS. Finally, the allelic variant of 308 bp of the marker P5 was positively correlated with IS, ID1, IW and IA, respectively, in decrescent order, and negatively correlated with IH (Table 3).

On the basis of the correlation observed among the inflorescence descriptors and the alleles detected for the five primers utilized we directed our attention to the alleles most correlated with at least four correlations with the six inflorescence descriptors utilized. The most correlated alleles chosen were the following: P1_155, P2_153, P2_157, P2_162, P2_165, P2_168, P3_184, P3_186, P3_190, P3_192, P4_288, P4_291, P5_294, P5_304 and P5_308 (Table 3 and Table 4).

Utilizing the data from the above cited allelic variants for the five primers utilized and for the six inflorescence descriptors, we established the related PCA, for which the main component (PC1) is positively correlated with ID1, IA, IW, P4_288, P1_155 and ID2, respectively, in decrescent order, whereas it was negatively correlated with P4_291, IH and P3_184 and represented 32.60 % of the total variance (Table 4, Figure 1a). With regard to the second principal component (PC2), it was positively correlated with P5_294 and negatively correlated with IS, and it represented 11.57% of the total variance (Table 4, Figure 1a). Concerning the third component (PC3), it was positively correlated with P3_190 and negatively correlated with P5_304, and it represented 9.68% of the total variance (Table 4, Figure 1a).

Based on the correlation and the PCA observed and to better discriminate the six inflorescence morphotypes studied, we chose among the 20 alleles detected 5 of them correlated with at least 4 of the 6 inflorescence descriptors utilized. The most correlated alleles chosen were P1_155, P2_165, P3_184, P4_288 and P5_308 (Table 3 and Table 4).

The PCA analysis performed on the highest correlated alleles with the inflorescence descriptors showed the PC1 positively correlated that ID1, IA, IW, P1_155, ID2, P4_288, P2_165 and P2_192, and negatively correlated with P4_291, P3_184 and IH, representing 49.80% of the total variance (Table 5, Figure 1b). Concerning the PC2, it was positively correlated to IS and negatively correlated to ID2, and it represented 15.29% of the total variance (Table 5, Figure 1b).

The PCA plot established by the 15 chosen alleles showed the genotypes studied distributed in three main groups (Figure 1a). The first group (A) is represented by the CWRs characterized by high value of IH and low values of IW and IA (Figure 1a). The second group (B) is represented by the broccoli genotypes distinguishable by high IS values and by the intermediate values of IH, IW, ID1, ID2 and IA (Figure 1a). Group C, instead, is represented by cauliflower genotypes followed by the broccoli F1 hybrids showing the highest values for IW, IH, ID2, ID1 and IA and the lowest for IS (Figure 1a). The PCA plot performed utilizing the most correlated allele for each primer, confirmed the three groups observed earlier but distinguished them better (Figure 1a). Group A is represented by all the *B. oleracea* complex species (n = 9), group B by the broccoli landraces and hybrids F1, and group C by the cauliflower landraces and hybrids F1, validating the efficiency of the five alleles and of the SSRs utilized to distinguish among *B. oleracea* crops and complex species (n = 9) (Figure 1b).

The PCA obtained utilizing the three highest correlated allelic variances is shown in Figure 2. In fact, the allelic variances P1_155, P2_165 and P4_288, which show the highest correlation with the examined bio-morphometric traits allowing the genotypes distribution in different clusters, are each represented by the different inflorescence morphotypes studied (Figure 2).

## 3. Discussion

*B*. *oleracea* species includes many important vegetable crops exhibiting high morphological diversity among them and their cultivars. In our work, the main inflorescence morphometric traits (IW, IH, ID1, ID2, IS and IA) allow us to distinguish among the *B*. *oleracea* inflorescence morphotypes, in accordance with Branca et al. [32] and Treccarichi et al. [20]. The plant materials were selected from the *B*. *oleracea* core and the *Brassica* wild relatives species (n = 9) collection of the Di3A of the University of Catania to individuate the morphometric and genetic diversity of the inflorescence just before the anthesis stage. Broccoli landraces showed low values of IW due to how they were traditionally consumed, which was focused on the consumption of the small elongated primary inflorescence having small tender and sweet leaves [32,36]. As confirmed by the bio-morphometric and molecular analysis performed in the present work and by several additional authors, the Sicilian broccoli and cauliflower landraces are well differentiated from each other and from the F1 hybrids [37]. In general, broccoli F1 hybrids resemble the cauliflower inflorescence architecture that is clearly differentiated by its huge hypertrophic inflorescence and wide angle of curvature. As reported by several authors, in fact, the allelic distribution of BoCAL and BoAP1 also have contributed to the diversification process of the Calabrese broccoli and of the cauliflower purple type, which is typical of the northeast side of Sicily [16,29].

*B. oleracea* wild relatives (n = 9), furthermore, have differential traits from the *B. oleracea* crops that can be improved for their resistance to biotic and abiotic stresses and to improve organoleptic and nutraceutical properties for enhancing the bioactive compound amount and profile by assessing and exploiting their genetic diversity [38,39]. The *B. oleracea* complex species (n = 9) utilized in our work are diploid species and coexist along the Sicilian and the genetic flux among them and with different *B*. *oleracea* crops and landraces was ascertained [38].

MADS box genes are differentially conserved in the *Brassica* genome, and their differential expression on the different *B. oleracea* crops and organs are responsible for the flower induction and for the inflorescence development. The functional characterization of the following genes was performed by Sheng et al. [33], highlighting their different expression patterns and the molecular regulation of the flower development.

In our previous work, we already detected for each SSR locus different numbers of alleles among the accessions and the inflorescence morphotypes studied; BoAP1 (P1) showed 12 alleles, BoTHL1 (P2) 8 alleles, BoABI1 (P3) 9 alleles, BoPLD1 (P4) 6 alleles, and PBCGSSRBo39 (SP5) 39 11 alleles, in accordance with Branca et al. [32] and Treccarichi et al. [20]. Several of the following alleles, were unconsciously selected and maintained by the growers selected for the size of the hypertrophic inflorescence and probably they were also introgressed by the genetic flux among the *B. oleracea* wild relatives (n = 9) and the first domesticated kales and sprouting broccoli landraces [13]. The correlation among the allelic variants and the inflorescence bio-morphometric traits showed that they increase in terms of value when BoPLD1 (P4) *locus* tends to heterozygosity. In reality we have observed the P4_288 allele which is homozygous or heterozygous for broccoli and cauliflower whereas for all the *B. oleracea* complex species (n = 9), except for one of the two *B. incana* studied (BY2), it is absent (Figure 1) [20].

In the work of Tonguç and Griffith [30], the molecular markers P1, P2, P3 and P4 were characterized and identified as candidate markers to assess genetic similarity in broccoli, cabbage and cauliflower, and they showed the polymorphism information content (PIC) value of 0.70, 0.60, 0.58 and 0.45 for P3, P4, P2 and P1, respectively. For the BoAP1 (P1) the allele P1_155 is generally heterozygous for broccoli and cauliflower, whereas for all the *B. oleracea* complex species (n = 9), except for one population each of *B. incana* (BY1) and *B. rupestris* (BU4) studied, it is absent (Figure 1). Regarding BoTHL1 (P2) the allele P2_165 generally expresses a heterozygous condition for broccoli and cauliflower, and it was absent for all *B. oleracea* complex species (n = 9), except for one *B. rupestris* studied (BU4), is absent (Figure 1). For the BoAB1 (P3) the allelic variants P3_184 is always absent for broccoli and cauliflower, whereas for *B. oleracea* complex species (n = 9), it was homozygous for two populations of *B. rupestris* (BU1, BU4) and for two populations of *B. icanca* (BY1, BY2) (Figure 1).

With regard to P5, it was developed and characterized by Burgess et al. [31] in silico by genome shotgun sequences and showed the highest PIC which was 0.83. In fact, we detected the allele P5_308 which was generally homozygous in cauliflower and broccoli landraces and absent for all the *B. oleracea* complex species (n = 9), except for one of the four *B. rupestris* (BU4), which in previous studies seems to be an escape population, is absent.

The high number of allelic variants individuated in our previous study confirmed that the following molecular markers, can be exploited for the construction of a genetic map with the different annotation related to the polymorphic loci and for the identification of diploid and amphiploid *Brassica* taxa. The following molecular markers also allowed us to perform a hierarchical clustering dendrogram distinguishing both broccoli and cauliflower landraces and F1 hybrids, and their crosses, respectively, in each different phylogenetic clade [32].

Noteworthy, for all the primers selected, the broccoli landrace BR9 and the cauliflower F1 hybrid CVF1.2 were isolated from the morphotype cluster for their distinctive features, such as the slender and the compact, huge inflorescence for BR9 and CVF1.2, respectively (Figure 2). Herein, we are providing more information about the MADS box domain allelic distribution and diversity focusing on the ones strictly related to the inflorescence traits. The data discussed will be utilized shortly for validating them by the GBS dataset in progress in the frame of the genotyping activities of the EU H2020 BRESOV project.

On the other hand, the alleles individuated can already be a solid base for using them for selecting progenies by MAS for hypertrophic inflorescence and size for organic breeding of broccoli and cauliflower and for establishing new organic heterogenous materials requested by the EU Directive 848/2018.

## 4. Materials and Methods

### 4.1. Plant Material

Plant material includes 31 accessions of Sicilian landraces of broccoli (*B. oleracea* var. *italica*) and cauliflower (*B. oleracea* var. *botrytis*) and 8 *B. oleracea* complex species population (crop wild relatives—CWRs) belonging to the *Brassica* active collection of the Department of Agriculture, Food and Environment (Di3A) of the University of Catania (UNICT), as shown in Table 6 and Figure 3. The plants were transplanted in an open field by block randomized experimental design, as described by Branca et al. [32]. Plants were characterized by their agronomical traits related to the inflorescence production, following the International Board for Plant Genetic Resources (IBPGR) descriptors. Examined traits include (IW), height (IH), diameter (ID1), shape (IS), angle of curvature (IA) and inflorescence stem thickness (ID2) and were analyzed by the laboratory of Biotechnology of Vegetable and Flower Crops of the Di3A department of the University of Catania (UNICT).

IW was calculated using an analytical scale, while the IH (cm) and ID1 (cm) traits were calculated using a meter rule, and ID2 (mm) was noted using a calibre. The IS descriptor represents the ratio between IH and ID1, while curvature angle IA (°) is the angle limited by the central vertical axes and the tangent one at the extreme part of the inflorescence, and it was calculated using goniometer.

### 4.2. DNA Extraction and PCR

DNA extraction was performed using the kit GenEluteTM Plant Genomic DNA Miniprep (Sigma Aldrich Inc., St. Louis, MI, USA) and 200 ng μL^−1^ were used for PCR reaction, as reported by Branca et al. [32]. PCRs were done using the primers list reported in Table 7, obtaining the flanking SSRs sequences by Tonguç and Griffiths [30] for BoTHL1, BoAP1, BoPLD1, and BoABI1 and by Burgess et al. [31]. SSRs genome allocation were checked using the basic local alignment search tool (BLAST) (version 1.17) and Ensembl database (release 2021, version 3) and Uniprot database (release 2021, version 3) was used to study encoding regions close to the gene of interest. DNA amplification was performed in a Perkin Elmer 9700 thermocycler (ABI, Foster City, CA, USA) as reported by Branca et al. [40]. Capillary electrophoresis was carried out by ABI PRISM 3130 Genetic 191 Analyser (Applied Biosystems, Waltham, MA, USA) as described by Branca et al. [32,37] and GeneMapper 3.7 software (Applied Biosystems, Waltham, MA, USA) was used to note the fragments size manually checking each alleles peak.

### 4.3. Data Analysis

The Allelic data set was codified by numeric scores, distinguished from 0 (absence of any allele), 1 (heterozygosity), 2 (homozygosity). The matrix generated from the following annotations was used for the sub-mentioned statistical analysis and is available in the H2020 BRESOV repository on the Zenodo database and is also present in the Appendix A in Appendix A. The Statistical analysis was performed using the SPSS software version 27. Data were transformed using the percentage rank of the analyzed matrix and normalized using the angular coefficient (DEGRES(ASIN(RACINE(x/100))). Pearson’s correlation was performed to identify the allelic variants involved in the size of inflorescence. The alleles that showed the highest correlation with the morphometric traits were selected. Moreover, the principal component analysis (PCA), as a powerful tool for clustering and dimension reduction, was also performed to discriminate the accessions studied and explain the variability among genotypes by the main components reducing the size of data by the factorial analysis regression method.

## 5. Conclusions

Genotyping techniques based on molecular markers can be useful for improving knowledge about putative genes controlled by quantitative loci regulating several complex traits such as the inflorescence size. Based on the achieved results, the allelic variants P1_155, P2_165 and P4_288 of the markers BoAP1, BoTHL1 and BoPLD1, respectively, were the most associated with the increase of inflorescence size, and they also facilitate genotype distribution into several clusters by Principal Component Analysis (PCA), represented by each different inflorescence morphotype studied. These three selected alleles could be utilized as molecular markers for organic breeding programs by molecular assisted selection (MAS), and they could be helpful to individuate progenies with hypertrophic inflorescence after crossing broccoli lines and cauliflower with *B. oleracea* wild relatives (n = 9) for transferring useful forgotten alleles, during the domestication process, for increasing biotic and abiotic stress resistance and for organoleptic, nutritional and nutraceutical traits. Of course, the matrix utilized will soon be compared with the new GBS dataset that will permit us to finely validate our present work highlighting the several mutations responsible of the hypertrophic inflorescence of *B. oleracea*. The molecular markers individuated which could be used for the fast selection of a new resilient, efficient and sustainable cultivar exploiting the wild ancestor of *Brassica oleracea* crops.

## Figures and Tables

**Figure 1 plants-12-00407-f001:**
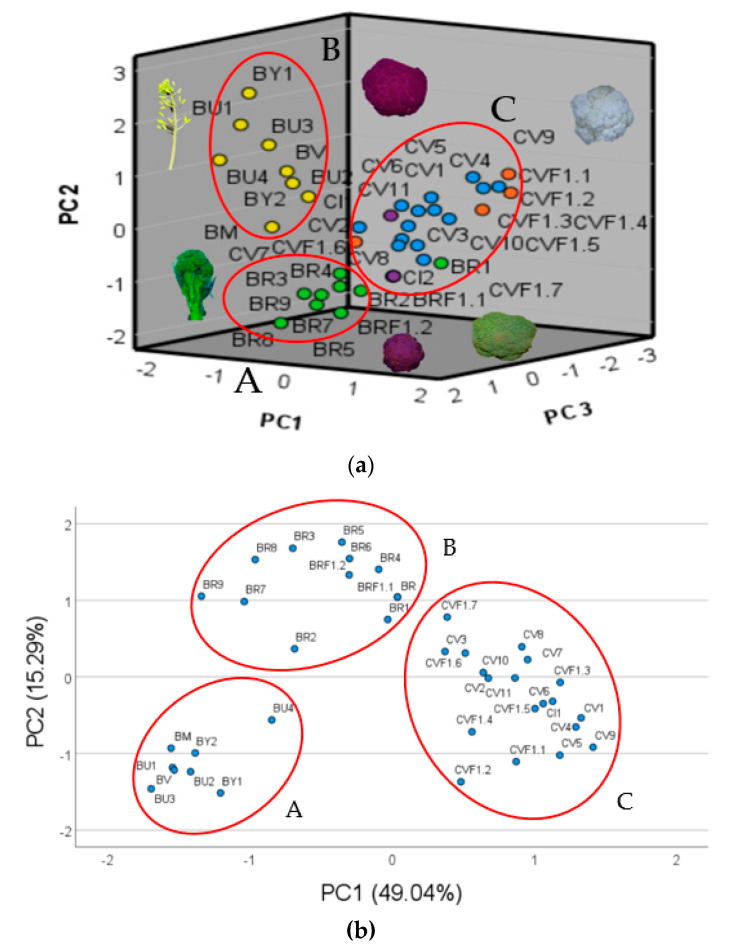
PCA plot performed utilizing 20 alleles selected for the primers utilized and all the inflorescence descriptors (**a**) and PCA plot performed utilizing the 5 more highly correlated alleles with the inflorescence traits (**b**), respectively.

**Figure 2 plants-12-00407-f002:**
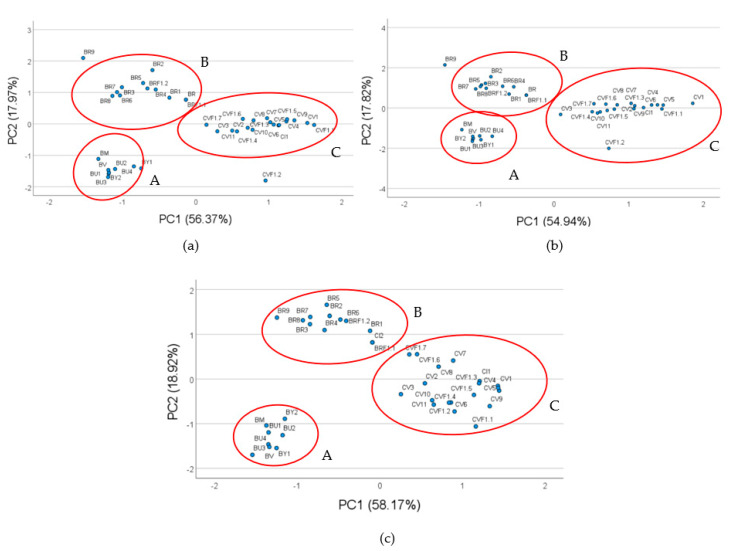
PCA plots performed utilizing three of the most correlated alleles with the inflorescence traits: allele of 155 bp for P1 (**a**), allele of 165 bp for P2 (**b**), allele of 288 bp for P4 (**c**), respectively.

**Figure 3 plants-12-00407-f003:**
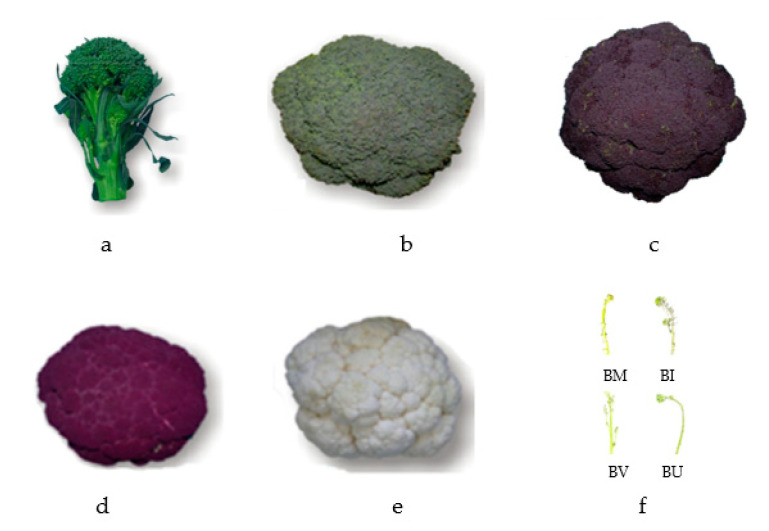
Inflorescence induction in relation of the different morphotypes tested which are, in the following order, (**a**): BR5, (**b**): BRF1.1, (**c**): BR41, (**d**): CV10, (**e**): CVF1.1, (**f**): BM1, BI1, BV and BU1, respectively.

**Table 1 plants-12-00407-t001:** Inflorescence morphometric traits analyzed.

Accession	IW	IH	ID2	ID1	IS	IA
CVF1.1	1095.8 (21.1) *	11.1 (8.4)	42.3 (8.5)	18.0 (8.7)	0.6 (9.6)	110.0 (21.9)
CV1	965.7 (37.4)	15.4 (14.6)	39.8 (16.4)	20.7 (17.4)	0.7 (16.6)	105.0 (19.4)
CV4	666.6 (42.5)	15.2 (13.2)	34.1 (19.6)	21.1 (15.0)	0.7 (12.1)	112.0 (20.4)
CI1	628.8 (33.7)	16.8 (16.6)	38.1 (18.5)	19.7 (14.6)	0.9 (14.6)	101.0 (22.5)
CVF1.2	605.0 (33.8)	8.9 (16.7)	31.0(10.3)	16.9 (11.8)	0.5 (12.1)	113.0 (13.3)
CV5	567.3 (38.2)	14.5 (15.6)	37.0 (19.8)	19.5 (13.1)	0.7 (17.29)	113.0 (13.5)
CV6	564.9 (37.0)	14.5 (20.7)	34.6 (12.6)	20 (15.1)	0.7 (18.7)	104.0 (16.7)
CV7	554.5 (56.7)	18.8 (20.4)	30.8 (26.9)	19.5 (19.3)	0.9 (29.8)	107.0 (17.7)
CVF1.3	541.5 (54.7)	13.7 (24.4)	32.3 (21.9)	18.9 (29.6)	0.7 (18.3)	112.0 (22.3)
CV8	503.9 (35.4)	16.8 (28.4)	32.4 (18.1)	16.5 (17.9)	1.0 (34.4)	100.0 (27.4)
CVF1.4	467.1 (41.1)	7.5 (20.9)	30.0 (13.3)	14.6 (15.7)	0.5 (11.1)	101.0 (15.6)
CVF1.5	461.8 (47.1)	10.8 (16.1)	33.0 (13.9)	17.5 (16.4)	0.6 (17.4)	110.0 (19.3)
CV9	453.5 (49.7)	11 (17.2)	35.6 (17.9)	18.1 (27.3)	0.6 (22.5)	117.0 (15.8)
CV10	443 (55.9)	12.7 (23.0)	36.4 (24.2)	16.7 (23.8)	0.8 (32.4)	91.0 (26.5)
CVF1.6	438.8 (84.4)	17.6 (24.4)	28.8 (28.1)	16.8 (29.4)	1.1 (34.2)	93.0 (17.7)
CI2	378.3 (46.2)	10.2 (21.0)	36.8 (16.9)	17.2 (19.5)	0.6 (17.8)	113.0 (17.5)
BRF1.1	319.8 (40.9)	14.1 (26.8)	3.5 (19.5)	12.3 (26.7)	1.2 (44.4)	76.0 (29.8)
CVF1.7	317.4 (42.0)	17.2 (22.2)	29.2 (28.1)	14.8 (16.0)	1.2 (33.1)	98.0 (21.6)
CV11	305.7 (68.2)	8.7 (20.7)	31.7 (18.1)	15.4 (22.8)	0.6 (19.7)	92.0 (25.2)
BR1	279 (39.0)	16.6 (18.1)	3.8 (17.3)	11.1 (23.7)	1.5 (28.2)	57.0 (21.5)
BR2	266.9 (33.4)	22.2 (30.9)	3.2 (13.2)	8.5 (32.7)	2.7 (37.4)	58.0 (19.4)
CV2	263.6 (56.1)	11.2 (28.3)	34.2 (18.8)	14.4 (22.0)	0.8 (21.2)	91.0 (23.4)
BR3	226.4 (39.6)	18.2 (12.9)	3.1 (26.8)	7.9 (29.4)	2.3 (30.5)	49.0 (27.8)
BR4	217.7 (58.3)	18.2 (18.2)	2.9 (29.8)	9.5 (31.6)	1.9 (29.4)	54.0 (26.3)
BRF1.2	212.8 (36.3)	12.8 (12.2)	3.1 (15.0)	7. 8 (23.1)	1.9 (16.5)	46.0 (24.1)
BR5	188.3 (51.8)	16.6 (23.4)	2.9 (24.3)	7.7 (28.3)	2.2 (24.2)	46.0 (24.1)
CV3	186.6 (41.3)	8.4 (17.5)	28.6 (16.7)	13.6 (15.1)	0.6 (18.2)	85.0 (24.8)
BR6	164.0 (49.0)	16.5 (17.9)	3.3 (32.4)	8.3 (29.5)	2.0 (52.3)	46.0 (32.8)
BR7	143.9 (42.2)	16.0(29.0)	2.7 (22.7)	7.8 (29.0)	2.1 (22.6)	48.0 (26.7)
BR8	109.5 (30.8)	15.5 (9.5)	2.6 (20.2)	7.9 (25.8)	2.0 (23.4)	41.0 (34.2)
BR9	63.1 (41.7)	16.9 (23.5)	2.7 (18.9)	4.7 (22.3)	3.6 (15.5)	27.0 (15.2)
BU1	33.3 (28.3)	27.6 (15.5)	16.2 (20.2)	3.1 (17.9)	0.2 (21.2)	14.0 (11.7)
BU2	28.7 (1.6)	19.5(1.5)	19.3 (3.3)	4.1 (0.2)	0.21 (0.1)	15.0 (0.9)
BY1	27.7 (3.7)	20.4 (1.0)	22.5 (4.5)	3.3(0.7)	0.1 (0.1)	13.5 (2.1)
BM	26.6 (5.9)	16.7 (4.6)	9.6 (2.5)	2.7 (0.4)	0.3 (0.1)	11.3 (2.6)
BU3	22.4 (0.4)	23.5 (4.0)	19.6 (1.5)	2.0 (0.3)	0.1 (0.1)	9.5 (0.7)
BU4	21.1 (0.8)	19.2 (2.2)	18.9(3.6)	2.2(0.1)	0.1 (0.1)	13.5 (2.1)
BY2	20.6 (1.3)	20.8 (0.6)	18.5 (1.6)	2.8 (0.4)	0.2 (0.1)	11.5 (0.7)
BV	19.7 (0.6)	14.8 (0.4)	19.0 (1.2)	2.4 (0.2)	0.1 (0.1)	10.5 (0.7)

numbers in brackets () * indicate the standard deviation.

**Table 2 plants-12-00407-t002:** Pearson’s correlation among traits.

Genotype	IW	IH	ID2	ID1	IS	IA
IW	1					
IH	0.024	1				
ID2	0.680 **	−0.035	1			
ID1	0.880 **	−0.066	0.724 **	1		
IS	−0.117	−0.068	−0.638 **	−0.107	1	
IA	0.847 **	−0.033	0.706 **	0.980 **	−0.086	1

** indicates that the correlation is significant at *p* < 0.01.

**Table 3 plants-12-00407-t003:** Correlation among all the allelic variants detected by the molecular markers used and the analyzed traits to individuate the most associated alleles of the examined traits.

Allelic Variant	IW	IH	ID2	ID1	IS	IA
P1_155	0.622 **	−0.471 **	0.521 **	0.622 **	0.032	0.677 **
P1_156	−0.101	0.156	0.219	−0.097	0.202	−0.135
P1_164	−0.375	0.072	−0.082	−0.334	−0.283	−0.306
P2_153	−0.288	0.189	0.219	0.308	0.00	−0.264
P2_157	−0.338 *	0.405 **	−0.088	−0.372 *	−0.376 *	−0.372 *
P2_162	−0.152	−0.029	−0.418 **	−0.266	0.196	−0.175
P2_165	−0.461 *	−0.220	0.583 **	0.594 **	−0.014	0.538 **
P2_168	0.160	0.021	0.226	0.205	0.050	0.204
P3_180	0.010	0.033	0.069	0.046	−0.003	0.095
P3_184	−0.455 **	0.440 **	0.123	−0.455 **	−0.477 **	−0.433 *
P3_186	−0.233	0.296	−0.214	−0.257	0.062	−0.187
P3_190	0.257	−0.440 *	0.268	0.303	0.192	0.226
P3_192	0.418 *	−0.324	0.222	0.424 **	0.156	0.436 **
P3_194	0.140	−0.015	−0.068	0.068	0.146	0.174
P4_282	−0.139	0.199	−0.097	−0.184	−0.168	−0.232
P4_288	0.460 **	−0.333 *	0.308	0.522 **	0.172	0.568 **
P4_291	−0.462 **	0.381 *	−0.462 **	−0.477 **	0.148	0.485 **
P5_294	−0.343 *	0.410 **	0.078	−0.376 *	−0.391 *	0.384 *
P5_304	0.306	0.050	0.089	0.217	0.165	0.330 *
P5_308	0.384 *	−0.474 **	0.132	0.449 **	0.478 **	0.380 *

* and ** indicate that the correlation is significant at *p* < 0.05 and *p* < 0.01, respectively.

**Table 4 plants-12-00407-t004:** Principal component of the rank of all the examined traits and for the most correlated allelic variants detected.

	PC1	PC2	PC3
IW	0.900	0.093	−0.155
IH	−0.564	0.108	0.132
ID2	0.670	0.653	−0.045
ID1	0.938	0.135	−0.049
IS	0.200	−0.858	0.147
IA	0.919	0.131	−0.213
P1_155	0.671	0.197	−0.112
P2_153	−0.350	−0.035	0.067
P2_157	−0.482	0.422	0.074
P2_162	−0.183	−0.396	−0.411
P2_165	0.579	0.283	0.462
P2_168	−0.269	−0.041	−0.041
P3_184	−0.570	0.503	0.089
P3_186	−0.353	−0.196	−0.144
P3_190	0.323	−0.085	0.693
P3_192	0.509	−0.150	−0.287
P4_288	0.673	−0.066	0.098
P4_291	−0.643	−0.162	0.031
P5_294	−0.503	0.523	0.027
P5_304	0.226	−0.046	−0.765
P5_308	0.526	−0.361	0.571
Variance (%)	32.60	11.57	9.68

**Table 5 plants-12-00407-t005:** The PCs matrix related to the bio-morphometric analysis and the selected allelic variants.

	PC1	PC2
IW	0.910	−0.058
IH	−0.554	−0.119
ID2	0.741	−0.595
ID1	0.952	−0.060
IS	0.101	0.888
IA	0.941	−0.100
P1_155	0.752	−0.078
P2_165	0.631	−0.031
P3_192	0.506	0.258
P4_288	0.644	0.156
P5_308	0.524	0.641
Variance (%)	*49.08*	*15.29*

**Table 6 plants-12-00407-t006:** List of *B. oleracea* complex species (*n* = 9) utilized in the experiment, with cauliflowers and broccoli F1 and landraces, respectively, and crop wild relatives.

Accession Code	Laboratory Code	Origin	Species
UNICT 583	BR 46	Vittoria	BR1
UNICT 658	BR 45 S1	Acireale	BR2
UNICT 658	BR 129	Roccella Valdemone	BR3
UNICT 657	BR 128	Roccella Valdemone	BR4
UNICT 655	BR 126	Adrano	BR5
UNICT 637	BR 106	Cefalù	BR6
UNICT 3675	BR 94 S1	Francavilla	BR7
UNICT 3668	BR 115 S1	Troina	BR8
UNICT 574	BR 36	Biancavilla	BR9
UNICT 3578	BR 165 Marathon	Esasem	BRF1.1
UNICT 651	BR 122 Packman	Petoseed	BRF1.2
UNICT 4145	BR 13 S3 AC	Modica	CI1
UNICT 579	BR 41	Modica	CI2
UNICT 3190	BR 15 S 1 A	Modica	CV1
UNICT 3669	BR 17 S2	Ragusa	CV2
UNICT 3674	CV 19 S2 A	Piazza Armerina	CV3
UNICT 4137	CV 99 S2 B	Adrano	CV4
UNICT 4138	CV 76 S2	Acireale	CV5
UNICT 3652	CV 159	Catania	CV6
UNICT 3900	BR 13 A X CV98/21	Di3A	CV7
UNICT 3895	CV 98/2 X CV 136 EG	Di3A	CV8
UNICT 3089	CV 75 S3AC	Acireale	CV9
UNICT 3906	CV 24 S4	Biancavilla	CV10
UNICT 3671	CV 72 S2	Catania	CV11
UNICT 3876	CV 171 Menhir F1	ISI sementi	CVF1.1
UNICT 3878	CV 173 Freedom	3878 Royal Sluis	CVF1.2
UNICT 3902	CV 33 S1	Royal Sluis	CVF1.3
UNICT 3880	CV 175 White Flash	Sakata	CVF1.4
UNICT 3879	CV 174 Graffiti	ISI sementi	CVF1.5
UNICT 3892	CV 98/2 X BR 13 S3	DISPA 3	CVF1.6
UNICT 3893	CV 136 EG X CV98/2	DISPA 1	CVF1.7
UNICT 342	*Brassica macrocarpa 1*	Favignana	BM
UNICT 733	*Brassica rupestris 1*	San Vito Lo Capo	BU1
UNICT 3270	*Brassica rupestris 2*	Bivongi	BU2
UNICT 732	*Brassica rupestris 3*	Roccella Valdemone	BU3
UNICT 736	*Brassica rupestris 4*	Ragusa Ibla	BU4
UNICT 3040	*Brassica villosa 1*	Marianopoli	BV
UNICT 3512	*Brassica incana 1*	Agnone Bagni	BY1
UNICT 4158	*Brassica incana 2*	Sortino	BY2

Legend: CV—Cauliflower; CI—Ciurietti landrace; BR—Broccoli; BY—*B*. *incana*; BM—*B*. *macrocarpa*; BU—*B*. *rupestris*; BV—*B*. *villosa*.

**Table 7 plants-12-00407-t007:** List of primers utilized with their sequences and chromosome (C) position.

Name	SSR Motif	Primer Sequence (Forward, Reverse)	C	Position (from–to); (bp)	Code
BoAP1	(AT)_9-1_	GGAGGAACGACCTTGATT GCCAAAATATACTATGCGTCT	C6	33,883,667–33,887,357	P1
BoTHL1	(CTT)_7_	GCCAAGGAGGAAATCGAAG AAGTGTCAATAAGGCAACAAGG	C9	17,254,558–17,255,176	P2
BoABI1	(TC)_16_	TATCAGGGTTTCCTGGGTTG GTGAACAAGAAGAAAAGAGAGCC	C1	1,229,915,511–12,992,170	P3
BoPLD1	(CT)_7_(AT)_7-1_	GACCACCGACTCCGATCTC AGACAAGCAAAATGCAAGGAA	C5	46037340–46,037,606	P4
PBCGSSRBo39	[GGTCG]_4_	AACGCATCCATCCTCACTTC TAAACCAGCTCGTTCGGTTC	C7	50595248–50595537	P5

## Data Availability

The matrix data presented in this study are deposited on Kibana and Zenodo database repositories, and it will be available after the embargo period of one year foreseen in the BRESOV Consortium Agreement.

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
