# Peer review of "Molecular Markers for Detecting Inflorescence Size of Brassica oleracea L. Crops and B. oleracea Complex Species (n = 9) Useful for Breeding of Broccoli (B. oleracea var. italica) and Cauliflower (B. oleracea var. botrytis)"

_plants, 2023, doi:10.3390/plants12020407_

Round 1
Reviewer 1 Report
The manuscript “Molecular Markers for Detecting Inflorescence Size of Brassica oleracea L. Crops and B. oleracea Complex Species (n=9) Useful for Breeding of Broccoli (B. oleracea var. italica) and Cauliflower (B. oleracea var. botrytis)” by Treccarichi et al., reports the analysis of the main inflorescence morphometric characters and genotyping of five SSR markers on a collection of broccoli, cauliflower, kale and wild Brassica accessions.
Major comments:
1, the authors used only 5 SSR markers to associate the genetic variations to the agronomic traits. The number of markers is far from the needed for doing this work, especially nowadays the genotyping cost by sequencing is very cheap, and the molecular markers are easy to obtain.
2, the authors analyzed the main inflorescence morphometric characters for B. oleracea Complex Species. This maybe the main highlight of this study. But the author didn’t show any figure about the plants. If the pictures of the inflorescence morphometric characters provided, it will significantly improve the readability of the manuscript.
Minor comments
1, in abstract, line 20, Branca et al. (2018) and Treccarichi et al. (2018), the numerical matrix….
Abstract should not contain citation.
2, line 130-136, IW, IH, ID2,ID1, IA, etc, should provide the full noun when it first appears.
3, line 220, the molecular weight of 157 bp, molecular weight should be fragment length?
4, figure 2, there are two (a), and two (b)
5, line 486, Examined traits include (IW), ?
6, table 7, what is the Code meant?
7, line 557-575, Data Availability Statement, Acknowledgments, Appendix A, Appendix B, these sections should be replaced by materials form authors.
8, references should be carefully checked according the journal format.
Author Response
Dear Plants editorial Board, thanks for all the revision done that allow us to improve our work.
Reviewer 1
Major comments:
1- the authors used only 5 SSR markers to associate the genetic variations to the agronomic traits. The number of markers is far from the needed for doing this work, especially nowadays the genotyping cost by sequencing is very cheap, and the molecular markers are easy to obtain.
A: Thank for the suggestion; we utilized 5 SSRs because in preliminary previous works they were able to discriminate inflorescence morphotypes among Brassica oleracea crops and wild species. With this work we would like to highlight some key alleles that could be useful just from now for MAS. Of course, your suggestion is interesting and the sequencing work is in progress by using the recent GBS database established in the frame of the EU H2020 BRESOV project.
2, the authors analyzed the main inflorescence morphometric characters for B. oleracea Complex Species. This maybe the main highlight of this study. But the author didn’t show any figure about the plants. If the pictures of the inflorescence morphometric characters provided, it will significantly improve the readability of the manuscript.
A: Dear reviewer, thanks for the advice: we added the figure for each morphotype evaluated and now the work is better.
Minor comments
1- in abstract, line 20, Branca et al. (2018) and Treccarichi et al. (2018), the numerical matrix….
Abstract should not contain citation.
A: The following references were removed from the abstract, thanks for the correction.
2- line 130-136, IW, IH, ID2,ID1, IA, etc, should provide the full noun when it first appears.
A: Thank you for the advice, we added the whole descriptors name in the first section of results.
3- line 220, the molecular weight of 157 bp, molecular weight should be fragment length?
A: Yes, it corresponds to the fragment length.
4- figure 2, there are two (a), and two (b)
A: Thank you for the correction, the figure was corrected.
5- line 486, Examined traits include (IW), ?
A: Yes, the examined traits include the IW.
6- table 7, what is the Code meant?
A: It is the working code used for the SSRs primers by our team and is present in correspondence of the fragment’s molecular weight in Table 7.
7- line 557-575, Data Availability Statement, Acknowledgments, Appendix A, Appendix B, these sections should be replaced by materials form authors.
A: The above-mentioned sections were implemented.
8- references should be carefully checked according the journal format
A: references were checked and modified according to the journal format

Reviewer 2 Report
The authors of this article screened 31 accessions of Sicilian landraces of broccoli and cauliflower and 8 crop wild relatives for six morphological traits. They also screened these accessions using five previously published SSR markers and tried to associate these markers with the morphological traits.
Considering the importance of brassica species in food and nutrition, this work is important However, I have a couple of questions and suggestions:
The figures must be improved, axis titles are not visible.
The title of figure 1 is missing.
Define each morphological trait (IW, IH, ID2, ID1, IS, IA) at the first instance they are mentioned in the text. It will help readers if a cartoon/hypothetical figure explaining/depicting these traits is included in the manuscript.
The materials and methods section lacks a lot of information and needs to be modified. Please include more information on statistical analysis such as how PCA was performed, and how correlation among all the allelic variants was performed to detect most associated alleles to the examined trait.
The manuscript requires extensive English editing.
Author Response
Dear Plants editorial Board, thanks for all the revision done that allow us to improve our work.
Reviwer 2
The authors of this article screened 31 accessions of Sicilian landraces of broccoli and cauliflower and 8 crop wild relatives for six morphological traits. They also screened these accessions using five previously published SSR markers and tried to associate these markers with the morphological traits.
Considering the importance of brassica species in food and nutrition, this work is important However, I have a couple of questions and suggestions:
The figures must be improved, axis titles are not visible.
A: Thanks for all the suggestions done; Figure were improved as suggested but we are available for additional modifications.
2-The title of figure 1 is missing.
A: Title was inserted and many thanks for the advice.
3-Define each morphological trait (IW, IH, ID2, ID1, IS, IA) at the first instance they are mentioned in the text. It will help readers if a cartoon/hypothetical figure explaining/depicting these traits is included in the manuscript.
A: Descriptors were explained and written in the first section of results.
4-The materials and methods section lacks a lot of information and needs to be modified. Please include more information on statistical analysis such as how PCA was performed, and how correlation among all the allelic variants was performed to detect most associated alleles to the examined trait.
A: Additional informations were added to the M&M section.
5-The manuscript requires extensive English editing.
A: Extensive English editing was performed as recommended.

Round 2
Reviewer 1 Report
The authors answered all my comments.
Author Response
English language and style are checked carefully, the modification is tracked in the text (in green)

Reviewer 2 Report
Dear Authors,
The X and Y-Axis texts are still not visible in the figures. Also, materials and methods need more detailed information on statistical analysis, mainly how were they performed.
Author Response
Dear Plants editorial Board, thanks for all the revision done that allow us to improve our work.
we have improved the different part of the article as requested.
1-Comment: The X and Y-Axis texts are still not visible in the figures.
Answer Figure were improved as suggested, the title of the axis are more clear in the new version
2-Comment: materials and methods need more detailed information on statistical analysis, mainly how were they performed.
Answer Thanks for the suggestions, more details about the approach are added to the section of statistical analysis, we explained how we normalized the data and we clarified that the selection of the alleles is based on the correlation with the morphometric data.
Moderate English changes required
Answer English language and style are checked carefully, the modification is tracked in the text (in green)

Round 3
Reviewer 2 Report
Dear Authors, Thank you for addressing the comments.